# Can We Reduce the Diagnostic Burden of Sleep Disorders? A Single-Centre Study of Subjective and Objective Sleep-Related Diagnostic Parameters

**DOI:** 10.3390/medicina61050780

**Published:** 2025-04-23

**Authors:** Tadas Vanagas, Domantė Lipskytė, Jovita Tamošiūnaitė, Kęstutis Petrikonis, Evelina Pajėdienė

**Affiliations:** 1Neurology Department, Medical Academy, Lithuanian University of Health Sciences, 44307 Kaunas, Lithuania; kestutis.petrikonis@lsmu.lt (K.P.); evelina.pajediene@lsmu.lt (E.P.); 2Medical Faculty, Medical Academy, Lithuanian University of Health Sciences, 44307 Kaunas, Lithuania; domante.lipskyte@stud.lsmu.lt (D.L.); jovitatam3@gmail.com (J.T.)

**Keywords:** sleep, polysomnography, sleep disorders, sleep questionnaires

## Abstract

*Background and Objectives:* Sleep disorders are highly prevalent in society and require focused attention within healthcare systems. While patient history, reported complaints, and subjective sleep questionnaires can provide initial insights into potential sleep issues, polysomnography (PSG) remains the gold standard for diagnosing various sleep disorders. However, long waiting times for PSG appointments in many healthcare facilities pose challenges for timely diagnosis and treatment. This study aimed to evaluate the diagnostic value of subjective measures, including patient-reported parameters, in comparison to the objective findings of PSG. *Materials and Methods:* In this study, we retrospectively analysed the data from 562 patients who underwent clinical evaluation and PSG testing at the Hospital of Lithuanian University of Health Sciences Kaunas Clinics between 2018 and 2024. We report the diagnostic accuracy of different sleep questionnaires to detect various sleep disorders in our population. *Results:* We report the corresponding sensitivity and specificity values: the Epworth Sleepiness Scale (ESS)—73.2% and 44.1% for detecting severe obstructive sleep apnoea and 87.1% and 76.8% for detecting hypersomnia; the Insomnia Severity Index (ISI)—77.2% and 63.3% for detecting insomnia; the Berlin Questionnaire (BQ)—67.8% and 68.8% for detecting obstructive sleep apnoea; the Ullanlina Narcolepsy Scale (UNS)—84.4% and 58.9% for detecting hypersomnia; the Innsbruck REM Sleep Behaviour Disorder Inventory (RBD-I)—93.3% and 52.5% for detecting RBD; the REM Sleep Behaviour Disorder Single-Question Screen (RBD1Q)—73.3% and 81.0% for detecting RBD; and the Paris Arousal Disorder Severity Scale (PADSS)—57.5% and 90.5% for detecting parasomnia. *Conclusions:* When comparing our findings with the previous literature, we found that the screening tools generally demonstrated a slightly poorer performance in our population. However, our results suggest that certain individual questions from the comprehensive questionnaires may provide comparable diagnostic values, while reducing the patient burden. We propose a targeted screening approach that integrates fundamental clinical parameters, key screening questions, and selected validated questionnaires, enabling primary care and outpatient clinicians to more efficiently identify patients who may require referral for specialised sleep evaluation and treatment.

## 1. Introduction

The global burden of sleep disorders is increasing, yet insufficient attention is given to sleep-related health issues. Given that sleep constitutes nearly one-third of our lives, disturbances in sleep quality and duration can have significant consequences on overall health. A recent review analysing data from over 180,000 patients during the COVID-19 pandemic reported the prevalence of poor sleep quality at an astonishing 47.12%, short sleep duration at 40.81%, long sleep duration at 31.61%, and insomnia symptoms at 21.15% [1]. Also, sleep disturbance is related to deteriorating cardiovascular health, an increased risk of stroke, myocardial infarction, memory problems, dementia, and other ailments [2]. These findings highlight the critical need for enhanced efforts in the prevention, early identification, treatment, and rehabilitation of sleep disturbances to promote healthy aging and improve overall well-being.

Patients presenting with sleep-related complaints are typically classified into one or more common diagnostic categories, such as obstructive sleep apnoea (OSA), insomnia, narcolepsy, parasomnias, restless leg syndrome (RLS), periodic limb movement disorder (PLMD), and rapid eye movement (REM) sleep behaviour disorder (RBD). Pinpointing the exact diagnosis often requires additional objective evaluation, with polysomnography (PSG) serving as the gold standard for confirming sleep disorders. However, the long waiting times for PSG appointments in many healthcare facilities pose challenges for timely diagnosis and treatment.

Before PSG, patients are usually instructed to maintain a sleep diary, document daily sleep complaints, and complete various validated questionnaires that help preliminarily identify potential conditions, such as OSA, parasomnias, and RBD. Although these questionnaires are not obligatory, they provide clinicians with valuable supplementary information for interpreting the PSG results and refining the diagnostic process.

In our clinical practice, we employ a range of validated questionnaires, including the Epworth Sleepiness Scale (ESS), the Insomnia Severity Index (ISI), the Berlin Questionnaire (BQ), the Ullanlina Narcolepsy Scale (UNS), the Innsbruck REM Sleep Behaviour Disorder Inventory (RBD-I), the REM Sleep Behaviour Disorder Single-Question Screen (RBDQ1), the Paris Arousal Disorder Severity Scale (PADSS), and the Restless Leg Syndrome Rating Scale (RLSRS). While this comprehensive approach aims to capture the full spectrum of sleep disorders, patients frequently express concerns about the volume of questionnaires. Many report feeling overwhelmed and perceive overlapping content among the tools, which can lead to reduced motivation and compliance.

This feedback raises critical considerations about the clinical utility and efficiency of these instruments. It calls for the reassessment of whether all the questionnaires are equally valuable in contributing to the diagnostic process, or whether certain tools could be consolidated to enhance patients’ experience without compromising diagnostic accuracy. Additionally, it highlights the potential for developing a single, highly sensitive and specific questionnaire that could streamline the diagnostic workflow and potentially minimise the reliance on PSG for certain sleep disorders. Such efforts would improve both patient satisfaction and the efficiency of clinical workflows.

Our aim was to critically assess the performance of sleep questionnaires in everyday clinical practice, evaluate the sensitivity and specificity values for detecting various sleep disorders, and compare them with the literature results. In this study, we retrospectively analysed the data from over 500 patients who underwent clinical evaluation and polysomnography (PSG) at the Hospital of Lithuanian University of Health Sciences Kaunas Clinics, Neurology Department between 2018 and 2024. We present the literature analysis and distribution of various sleep disorders within our clinical setting, along with demographic characteristics and the most frequently observed comorbidities. Additionally, we evaluated the diagnostic accuracy of these commonly used sleep questionnaires and assessed their relevance in the diagnostic process.

### 1.1. Obstructive Sleep Apnoea

OSA is a sleep breathing disorder characterised by recurrent episodes of complete (apnoea) or partial (hypopnea) upper airway collapse during sleep, which result in intermittent oxygen desaturation and/or arousal from sleep, leading to sleep fragmentation or non-restorative sleep [3]. Individuals with suspected OSA commonly present with symptoms such as excessive daytime sleepiness, loud snoring, and episodes of gasping, choking, or apnoea during sleep, which are frequently reported by a bed partner [4,5]. PSG is considered the gold standard for the diagnosis of OSA. It is typically diagnosed when the apnoea–hypopnea index (AHI) or the respiratory disturbance index (RDI) is ≥15 events per hour or AHI/RDI ≥ 5 events/hour in the presence of one or more associated symptoms, such as excessive daytime sleepiness, fatigue, waking up with sensations of breath-holding, and loud snoring. OSA is classified as mild (AHI/RDI ≥ 5 events/hour), moderate (AHI/RDI ≥ 15 events/hour), or severe (AHI/RDI ≥ 30 events/hour) [6,7]. Alternative screening tools, including the Berlin Questionnaire (BQ), the STOP-BANG Questionnaire, the STOP Questionnaire, and the Epworth Sleepiness Scale (ESS), have been widely used for the evaluation of OSA [7].

The first screening tool for OSA which was included in our study is the Epworth Sleepiness Scale (ESS). The ESS is an eight-item questionnaire designed to assess daytime sleepiness in adults. It evaluates the propensity to fall asleep during routine activities, such as reading, watching television, and driving, by assigning a score from 0 to 3 for each activity. The total score ranges from 0 to 24, with higher values indicating greater levels of daytime sleepiness. While a score ≤ 10 is considered within the normal range, score > 10 suggests pathological sleepiness [8]. Among other questionnaires, the ESS has the lowest sensitivity for detecting OSA, with sensitivity values of 54% for mild OSA, 47% for moderate OSA, and 58% for severe OSA, and demonstrates relatively higher specificity, with values of 65% for mild OSA, 62% for moderate OSA, and 60% for severe OSA [7]. Despite its low sensitivity for detecting OSA, the ESS demonstrates acceptable test–retest reliability for evaluating treatment responses to stimulant therapy in clinical trial settings [9].

Another screening tool for OSA included in our study is the BQ. The BQ consists of questions grouped into three categories: category 1—snoring; category 2—daytime sleepiness; category 3—hypertension and body mass index (BMI). Scoring is determined as follows: categories 1 and 2 are considered positive if frequent symptoms (>3–4 times per week) are reported, while category 3 is positive if there is a history of hypertension or a BMI > 30 kg/m^2^. A positive score in two or more categories indicates a high risk of OSA, with a sensitivity of 86% and a specificity of 77% [10]. When compared to other questionnaires, the BQ demonstrates sensitivity rates of 76% for mild, 77% for moderate and 84% for severe OSA, along with specificity rates of 59% for mild, 44% for moderate, and 38% for severe OSA [7]. However, due to its low sensitivity and specificity, BQ is not typically used as a diagnostic tool for patients with OSA in sleep clinics [11].

Several alternative questionnaires not included in our study are also effective screening tools for OSA. Compared to the other questionnaires, the STOP-BANG has the highest sensitivity rates—88% for mild OSA, 90% for moderate OSA, and 93% for severe OSA—but also shows lower specificity rates of 42%, 36%, and 35% for mild, moderate, and severe OSA, respectively [7].

### 1.2. Insomnia

In our study, we also analysed another prevalent sleep disorder—insomnia. According to *International Classification of Sleep Disorders, Third Edition (ICSD-3)* [12], insomnia is defined as difficulty initiating or maintaining sleep or experiencing poor sleep quality, despite adequate sleep opportunities for sleep, leading to daytime dysfunction. Insomnia can be classified as either short-term or chronic. Short-term insomnia lasts a few days or weeks, while chronic insomnia occurs at least three times per week and persists for at least three months. The diagnosis of insomnia is confirmed when all four criteria are met: difficulty initiating or maintaining sleep; sleep difficulties occurring despite adequate opportunities to sleep; daytime dysfunction, which may include fatigue, malaise, concentration, or memory impairment and other related symptoms; and the sleep disorder is not caused by other medical conditions, medications, or substance use [12,13].

Self-report questionnaires are commonly used to evaluate the symptoms and severity of insomnia. In our study, we included the Insomnia Severity Index (ISI) questionnaire, which consists of seven items evaluating sleep disturbances, including sleep onset, sleep maintenance and early morning awakening problems, sleep dissatisfaction, daytime dysfunction, the perceived noticeability of sleep problems by others, and distress caused by insomnia. Each item is scored using a five-point Likert scale, ranging from 0 (no problem) to 4 (severe problem), with a total score from 0 to 28. The scoring thresholds classify individuals as follows: 0–7 (no clinically significant insomnia), 8–14 (sub-threshold insomnia), 15–21 (moderate insomnia), and 22–28 (severe insomnia). A total score ≥ 10 demonstrates a sensitivity of 86.1% and specificity of 87.7% for detecting insomnia in community samples [14].

### 1.3. Hypersomnia

Hypersomnia is a group of sleep disorders characterised by excessive daytime sleepiness (EDS) despite sufficient or even prolonged sleep duration, which greatly affects daily functioning. The main types of hypersomnia are narcolepsy (types I and II), idiopathic hypersomnia, Kleine–Levin syndrome, insufficient sleep syndrome, and secondary hypersomnia due to medical or psychiatric conditions (e.g., Parkinson’s disease, depression, multiple sclerosis, and traumatic brain injury) or due to medication and substance use (e.g., alcohol and drugs). The main disorder in this group is narcolepsy. It is a sleep disorder, which is characterised by clinical features, such as cataplexy, excessive daytime sleepiness, hypnagogic or hypnopompic hallucinations, sleep paralysis, and disrupted night-time sleep. According to the ICSD-S-TR criteria, a narcolepsy diagnosis is confirmed when the person experiences cataplexy and either has sleep latency of ≤8 min and two or more sleep-onset REM periods (SOREMPs) on a Multiple Sleep Latency Test (MSLT) [15], or there is an SOREMP (within 15 min of sleep onset) detected on a nocturnal polysomnogram. Narcolepsy type I is also diagnosed when they are deficient in hypocretin (≤110 pg/mL) [15]. However, various screening questionnaires are also useful for identifying narcolepsy symptoms [16,17,18]. The ESS, which evaluates excessive daytime sleepiness, is not only used for OSA screening, but is also highly relevant in narcolepsy evaluation. For patients with narcolepsy, the ESS scores are typically greater than 15 [17,18]. The Ullanlinna Narcolepsy Scale (UNS) included in our study is a validated questionnaire used for assessing the symptoms associated with narcolepsy. It consists of 11 items that evaluate the two main narcolepsy clinical symptoms—cataplexy and abnormal sleep patterns. The total score ranges between 0 and 44, with a mean score of 27.3 observed in patients with narcolepsy [16]. A score below nine strongly suggests against diagnosis [19]. The UNS demonstrates an excellent diagnostic performance, with a sensitivity of 100% and a specificity of 98.8% [16]. However, another study found that the sensitivity and the specificity for the UNS for differentiating narcolepsy type 1 from other disorders were 83.5% and 84.1%, respectively [19]. An alternative tool for the screening of narcolepsy is the Swiss narcolepsy scale (SNS), with a reported sensitivity of 89% and specificity of 88% [20].

### 1.4. Parasomnias

Parasomnias are sleep disorders involving abnormal, disruptive motor, verbal, or behavioural activities during sleep or transitions between wakefulness and sleep [21]. This group of disorders is categorised as non-REM (NREM), REM-related, and other parasomnias. NREM parasomnias include confusional arousals, sleepwalking, sleep terrors, sexsomnia, and sleep-related eating disorders. These conditions are linked to the N3 slow-wave stage of NREM sleep [22].

Effective diagnostic tools are crucial for distinguishing parasomnias from other conditions and providing proper treatment [23,24]. Diagnosis primarily relies on clinical history, including sleep patterns, medication use, and input from bed partners or parents [25,26]. Video-PSG is a key diagnostic tool for NREM parasomnias and differential diagnostics for other causes of nocturnal paroxysms.

The Paris Arousal Disorder Severity Scale (PADSS) is a validated questionnaire for assessing parasomnia symptoms. It is self-administered in less than five minutes and has three sections: behaviours (PADSS-A), episode frequency (PADSS-B), and disorder impact (PADSS-C). Behaviours are scored from 0 to 2, episode frequency from “never” to “more than two per night”, and consequences from “never” to “often”. The total scores range from 0 to 50. Options like “less than one episode per year” ensure normal controls can report infrequent symptoms like night screaming. The PADSS shows strong diagnostic accuracy, with a sensitivity of 83.6% and specificity of 87.8% [27]. However, the PADSS was specifically designed for assessing sleepwalking and sleep terrors. A Dutch study has suggested further research to evaluate the questionnaire’s applicability across different countries and its ability to detect other NREM parasomnias [28].

The Arousal Disorders Questionnaire (ADQ) is an alternative to the PADSS for diagnosing confusional arousal, such as sleepwalking and sleep terrors. It has demonstrated a sensitivity of 83% and a specificity of 93%. The ADQ is useful in clinical settings, particularly sleep and epilepsy centres, as it reduces reliance on resource-intensive procedures like PSG. This makes it a practical tool for the early detection and management of parasomnias [29].

### 1.5. REM Sleep Behaviour Disorder (RBD)

RBD is an REM sleep parasomnia characterised by the loss of muscle atonia and the physical enactment of vivid, often aggressive dreams during the REM phase of sleep [30,31].

The early diagnosis of RBD is critical for preventing injuries caused by dream enactment behaviours and for identifying the early stages of neurodegenerative conditions, particularly synucleinopathies, including Parkinson’s disease, multiple system atrophy, and dementia with Lewy bodies [31,32,33]. The diagnostic process for RBD should adhere to the criteria outlined in the International Classification of Sleep Disorders, 3rd Edition (ICSD-3-TR). The initial steps include a comprehensive clinical evaluation and the identification of a positive history of RBD symptoms [34]. Subsequently, PSG is mandatory to confirm REM sleep atonia (RWA) and to document detailed motor behaviours and vocalizations during REM sleep. Before RBD diagnosis is established, other possible causes, including alternative sleep disorders, mental health conditions, medication effects, and substance use, must be excluded [33].

The RBD Single-Question Screen (RBD1Q) is an effective tool for rapid initial screening in primary care settings. It uses a single “yes” or “no” response to assess the presence of symptoms suggestive of RBD. According to a Canadian study, the RBD1Q demonstrated a robust performance, with a sensitivity of 93.8% and a specificity of 87.2%, aiding in the identification of patients requiring further evaluation [35].

For more comprehensive screening following the initial step, the Innsbruck REM Sleep Behaviour Disorder Inventory (RBD-I) is a reliable tool. This questionnaire includes seven REM-specific and two non-REM control items, providing a detailed evaluation of RBD-related symptoms, such as dream enactment, vocalizations, and motor activity. The results yield a numerical score that reflects the likelihood of RBD, with a cutoff score of 0.25, calculated as the ratio of positive symptoms to the total number of questions answered. The RBD-I demonstrated diagnostic accuracy, with a sensitivity of 91.4% and a specificity of 85.7% [36].

The RBD Screening Questionnaire (RBDSQ) is another validated alternative for identifying probable cases of RBD. It consists of 10 items evaluating symptoms such as dream enactment behaviour, specific dream content, complex motor activity during sleep, dream recall, sleep disturbances, and coexisting neurological conditions. The scores range from 0 to 13, with a cutoff value of 5 or higher, indicating a positive result [37]. The RBDSQ has been validated in multiple languages and settings. For example, the Italian version of the RBDSQ demonstrated a sensitivity of 84.2% [38]. Additionally, its specificity was reported to be 78%, affirming its reliability in detecting RBD [39].

## 2. Materials and Methods

Approval to conduct this study was obtained from the Kaunas Regional Biomedical Research Ethics Committee on 30 January 2025 (Approval No. BE-2-7). We retrospectively analysed the data from 562 adult patients who underwent routine video-polysomnography (PSG) at the Neurology Department, the Hospital of Lithuanian University of Health Sciences Kaunas Clinics, between 2018 and 2024. All the patients were referred for PSG due to subjective sleep disturbances identified during routine neurological consultations in the outpatient department. Our study’s inclusion criteria were adult patients (>18 years old), patients who gave answers to the specific sleep questionnaires (ESS, ISI, BQ, UNS, RBD-I, RBD1Q, and PADSS), and patients who underwent PSG testing. The sleep questionnaires used in this study were previously validated in the Lithuanian language. The exclusion criteria were children (<18 years of age), patients who did not fill in sleep questionnaires, patients who did not undergo PSG testing, and patients whose clinical data were missing. The inclusion process of patient cases is illustrated in the STROBE diagram (Figure 1). For each patient, we assessed paper-based records detailing sleep quality, comorbidities, regularly used medications, and responses to the validated sleep questionnaires. Additionally, we analysed the video-PSG results to establish PSG-confirmed diagnoses. To ensure diagnostic accuracy, final sleep diagnosis was verified by cross-referencing the PSG findings with the patients’ clinical records.

Statistical analysis was performed using the IBM SPSS 29.0 program. The data in the tables are presented using absolute and percentage values and mean ± standard deviation (SD). For the non-normally distributed data, the non-parametric Mann–Whitney U test was used. For the comparison of the categorical data, the Chi-Square (χ^2^) Test was used. To test the diagnostic accuracy of the different screening tools, a Receiver Operatic Characteristic (ROC) curve was used. The data from ROC curve analysis are expressed using the area under the curve (AUC), 95% confidence intervals (95% CIs), and sensitivity and specificity values at an optimal cutoff value. Statistically significant values were considered when *p* < 0.05.

## 3. Results

Out of 562 patients included in the final analysis, 362 (64.4%) were diagnosed with obstructive sleep apnoea (OSA), 80 (14.2%) with insomnia, 32 (5.7%) with hypersomnia, 40 (7.1%) with parasomnia, and 16 (2.8%) with REM sleep behaviour disorder (RBD). Seventy-four patients (12.6%) received more than one sleep disorder diagnosis. The demographic characteristics are presented in Table 1. Significant gender differences were observed in the distribution of OSA and hypersomnia, with males more frequently diagnosed with OSA (*p* = 0.015) and females more commonly affected by hypersomnia (*p* < 0.001). Age-related variations were also noted, as the patients with OSA (*p* < 0.001), insomnia (*p* = 0.006), and RBD (*p* = 0.003) were significantly older, whereas those with hypersomnia (*p* = 0.004) and parasomnia (*p* < 0.001) were significantly younger. The body mass index (BMI) was significantly higher in the patients with OSA (*p* < 0.001), but showed no substantial differences among the other sleep disorder groups. Additionally, the patients with OSA exhibited the highest burden of comorbidities, including primary arterial hypertension (PAH), atrial fibrillation (AF), diabetes, and dyslipidaemia, compared to those with the other sleep disorders.

### 3.1. Obstructive Sleep Apnoea (OSA)

The diagnosis of obstructive sleep apnoea (OSA) was based on the polysomnography (PSG) recordings using the apnoea–hypopnea index (AHI) as the defining criterion. Among the 562 patients evaluated over seven years in our sleep centre, 362 (64.4%) were diagnosed with OSA (AHI > 5/h). Within this group, mild OSA (AHI 5–15/h) was identified in 162 (44.8%) patients, moderate OSA (AHI 15–30/h) in 90 (24.9%), and severe OSA (AHI > 30/h) in 100 (27.6%) patients. The Berlin Questionnaire (BQ) was used for OSA screening, with 560 completed evaluations. Among the BQ categories, 341 (60.7%) patients reported habitual snoring (≥3–4 nights per week) or loud snoring disturbing their sleep partner, or witnessed apnoea symptoms (positive category 1). Excessive daytime sleepiness or fatigue (positive category 2) was reported by 252 (44.8%) patients, with some experiencing episodes of falling asleep while driving. Category 3, which assesses body mass index (BMI) and arterial hypertension, was positive in 295 (52.5%) patients. A high risk of OSA (≥2 positive BQ categories) was identified in 304 (54.1%) patients, while 256 (45.6%) were classified as low-risk. The overall diagnostic performance of the BQ in detecting any stage of OSA in our cohort was moderate, with a sensitivity of 67.6% and a specificity of 68.8% (AUC = 0.682, 95% CI 0.636–0.729, *p* < 0.001) based on ROC curve analysis. Notably, category 1 and category 3 individually demonstrated an improved diagnostic performance; category 1 showed a sensitivity of 75.1% and a specificity of 63.8% (AUC = 0.694, 95% CI 0.648–0.741, *p* < 0.001), while category 3 had a sensitivity of 68.6% and a specificity of 74.9% (AUC = 0.718, 95% CI 0.673–0.762, *p* < 0.001). Table 2 and Figure 2 summarise the overall performance of the BQ and its individual components in diagnosing mild, moderate, and severe OSA (only the statistically significant results are presented).

### 3.2. Insomnia

In our cohort, 80 patients (14.2%) were diagnosed with insomnia. The ISI score for the patients with insomnia was significantly higher (17.84 ± 4.94 vs. 12.24 ± 5.98, *p* < 0.001). The ROC curve analysis of the ISI for insomnia diagnosis demonstrated moderate sensitivity (77.2%) and specificity (63.3%) at an optimal cutoff of 14.5 points (AUC = 0.762, 95% CI 0.710–0.814, *p* < 0.001) (Figure 3A).

### 3.3. Parasomnia

Among the 562 cases, 40 patients (7.1%) were diagnosed with parasomnia, excluding the cases of REM sleep behaviour disorder (RBD), which were analysed separately. For screening, we routinely employed the Paris Arousal Disorder Severity Scale (PADSS) to assess the parasomnia risk. The PADSS score for the patients with parasomnia was significantly higher (13.35 ± 6.82 vs. 5.7 ± 4.88, *p* < 0.001). The ROC curve analysis of the PADSS for parasomnia diagnosis demonstrated moderate sensitivity (57.5%), but high specificity (90.5%) at an optimal cutoff of 12.5 points (AUC = 0.822, 95% CI 0.753–0.801, *p* < 0.001) (Figure 3B).

### 3.4. Hypersomnia

Hypersomnia was diagnosed based on the PSG recordings, with the Multiple Sleep Latency Test (MSLT) performed when indicated. A total of 32 patients (5.7%) were diagnosed with idiopathic hypersomnia. The UNS score for the patients with hypersomnia was significantly higher (11.5 ± 7.07 vs. 5.71 ± 4.02, *p* < 0.001). The ROC curve analysis of the UNS for hypersomnia screening yielded moderate sensitivity (84.4%) and specificity (58.0%) at a cutoff of 5.5 points (AUC = 0.784, 95% CI 0.711–0.858, *p* < 0.001) (Figure 3C).

### 3.5. REM Sleep Behaviour Disorder (RBD)

Among the 562 participants, 16 (2.8%) were diagnosed with RBD. The RBD-I-9 score for the patients with RBD was significantly higher (0.38 ± 0.21 vs. 0.2 ± 0.19, *p* < 0.001). As shown in Figure 4, the sensitivity and specificity of the RBD1Q were 73.3% and 81.0%, respectively (AUC = 0.772, 95% CI 0.641–0.903, *p* < 0.001). The RBD-I-9 scale showed higher sensitivity (93.3%), but lower specificity (52.5%) at a cutoff value of 0.17 (AUC = 0.758, 95% CI 0.665–0.851, *p* = 0.001). These findings indicate that the RBD-I-9 is more effective in identifying RBD cases in clinical settings, while a negative RBDQ1 result is more reliable in ruling out the disorder. Additionally, we analysed individual items of the RBD-I-9 scale to determine their screening utility. Interestingly, questions 1 (sensitivity 75.0%; specificity 58.1%) and 2 (sensitivity 75.0%; specificity 84.2%) demonstrated the highest diagnostic value (Table 3). These questions assess whether the patient experiences dreams involving aggressive situations requiring self-defence and whether they scream, talk, or curse during sleep.

### 3.6. Epworth Sleepiness Scale (ESS)

Figure 5 represents diagnostic accuracy of the ESS in detecting different sleep disorders. The ESS showed statistically significant power only in detecting severe OSA (AUC 0.603, 95% CI 0.541–0.665, *p* = 0.001; sensitivity 73.2% and specificity of 44.1% at a cutoff value of 5.5 points) and hypersomnia (AUC 0.823, 95% CI 0.754–0.892, *p* < 0.001; sensitivity 87.1% and specificity 76.8% at a cutoff value of 10.5 points).

## 4. Discussion

In our clinical practice, we offer patients the opportunity to undergo video-polysomnography (video-PSG) testing, providing an evidence-based diagnosis along with tailored recommendations for further management. However, much can be achieved in the outpatient setting, particularly in primary care clinics, to facilitate the early detection and prevention of sleep disorders. While sleep questionnaires are valuable screening tools, they can be extensive and time-consuming, creating barriers to implementation in busy clinical settings.

Our study demonstrates that the widely used Epworth Sleepiness Scale (ESS) is only moderately effective in detecting severe obstructive sleep apnoea (OSA) (sensitivity 73.2%; specificity 44.1%), but performs better in detecting hypersomnia (sensitivity 87.1%; specificity 76.8%). These findings differ somewhat from the previous studies, which reported sensitivity and specificity values of 54% and 65% for mild OSA, 47% and 62% for moderate OSA, and 58% and 60% for severe OSA using the ESS [7]. Sample size was much higher in the citated meta-analysis—almost 50,000 patients. This discrepancy may reflect differences in our patient population or in the administration and interpretation of the questionnaire in our clinical context.

The Berlin Questionnaire (BQ) also showed a moderate overall performance (sensitivity 67.8%; specificity 68.8%), which is lower than the originally reported sensitivity of 86% and specificity of 77% (similar sample size—242 patients) [10]. Interestingly, we found that the positive category 3, which considers body mass index (BMI) and blood pressure, emerged as a simpler and potentially more effective screening tool for OSA risk (sensitivity 68.6%; specificity 74.9%). This finding suggests that basic clinical parameters easily obtainable in primary care settings may offer comparable screening utility to more complex questionnaires for certain sleep disorders.

When comparing our findings with the previous literature, we observed that our screening tools generally performed slightly worse in our population. For insomnia, the ISI showed a sensitivity of 77.2% and a specificity of 63.3% compared to the reported values of 86.1% and 87.7% in community samples (higher sample size—183 patients) [14]. For parasomnia, the PADSS demonstrated lower sensitivity (57.5%), but higher specificity (90.5%) compared to the originally reported sensitivity of 83.6% and specificity of 87.8% (higher sample size—73 patients) [27]. For hypersomnia, the UNS showed a sensitivity of 84.4% and a specificity of 58.0%, which differ from the better diagnostic performance originally reported in a larger-sample-size narcolepsy (267 patients) study—sensitivity 83.5% and specificity 84.2% [19]. These discrepancies highlight the importance of validating screening tools in diverse clinical settings and populations. Factors such as cultural and linguistic differences, comorbidity patterns, referral biases, and variations in questionnaire administration may all contribute to the observed differences. Additionally, the retrospective nature of our study and the specialised sleep centre setting may have influenced the patient composition and the questionnaire responses. Regarding REM sleep behaviour disorder (RBD), we found that the RBD1Q (sensitivity 73.3%; specificity 81.0%) was somewhat superior to the RBD-I-9 (sensitivity 93.3%; specificity 52.5%) in our population. These values differ from the previously reported figures for the RBD1Q (sensitivity 93.8%; specificity 87.2%; higher sample size—242 patients with idiopathic RBD) [35] and the RBD-I-9 (sensitivity 91.4%; specificity 85.7%; higher sample size—70 patients with RBD) [36]. Interestingly, our analysis revealed that question 2 of the RBD-I-9, which assesses dreams involving aggression and vocalizations during sleep, demonstrated an improved prognostic performance (sensitivity 75.0%; specificity 84.2%). This finding suggests that selected individual questions from comprehensive questionnaires might offer comparable diagnostic utility with reduced patient burden.

Based on our findings and their comparison with the existing literature, we propose an efficient screening approach for sleep complaints in outpatient settings. This approach should include the following:

1. Basic clinical assessment. The measurement of BMI and blood pressure, upper airway and mandibular anatomical features, along with a focused history of snoring and apnoeic episodes, particularly for OSA screening.

2. Targeted screening questions: For parasomnias, inquire about unusual nocturnal behaviours; for RBD, use the single-question RBD1Q (“Do you act out your dreams?”) or the RBD-I-9 question 2 (“Do you scream, talk, or curse during sleep?”).

3. Selected validated questionnaires: Apply the ESS for evaluating excessive daytime sleepiness, particularly when hypersomnia is suspected; use the ISI for insomnia symptoms; and employ the PADSS when parasomnia is suspected based on the initial screening questions.

This streamlined approach should cover the majority of common sleep disorders, including OSA, insomnia, parasomnias, hypersomnia, and RBD, while minimizing the questionnaire burden on patients and clinicians. The approach could be further refined by developing digital tools or decision support systems that guide clinicians through this targeted screening process based on the presenting symptoms.

Several limitations should be acknowledged in interpreting our findings. First, our study population consisted of patients referred to a specialised sleep centre, which may represent more severe or complex cases compared to the general population or those in primary care settings. Second, we did not specifically assess the questionnaire performance for restless leg syndrome (RLS) and periodic limb movement disorder (PLMD), which are important components of the sleep disorder spectrum. Third, cultural and linguistic factors may have influenced the questionnaire responses, as most of these tools were originally developed in different settings. Fourth, the retrospective design limited our ability to standardise questionnaire administration and interpretation across all the patients. Also, the reliability of our results could have been influenced by the small sample sizes, particularly for hypersomnia, parasomnia, and RBD. Finally, we did not evaluate the economic implications or patient acceptability of the different screening approaches.

Future research should focus on prospectively validating our proposed streamlined screening approach in diverse clinical settings, particularly in primary care, where early detection is most valuable. Additionally, the development and validation of culturally appropriate, brief screening tools that can simultaneously assess multiple sleep disorders would be beneficial. Studies examining the integration of subjective questionnaires with objective measures, such as at-home sleep testing or wearable technology, could further enhance diagnostic accuracy, while maintaining accessibility. Future studies would also benefit from exploring factors that influence the accuracy of subjective questionnaires in diagnosing various sleep disorders and analyse their potential application in clinical practice. Lastly, evaluating the long-term outcomes and cost-effectiveness of early detection through optimised screening protocols would provide valuable information for healthcare policy and resource allocation.

## 5. Conclusions

While our findings confirm that the validated sleep questionnaires have utility in screening for various sleep disorders, their performances in our clinical setting differed somewhat from the previously reported values. Our results support a more focused and strategic approach to sleep disorder screening, which could improve early detection, while optimizing resource utilization. By implementing a targeted screening protocol that combines the basic clinical parameters, the key screening questions, and selected validated questionnaires, clinicians in primary care and other outpatient settings could more effectively identify the patients who would benefit from referral for specialised sleep evaluation and treatment.

## Figures and Tables

**Figure 1 medicina-61-00780-f001:**
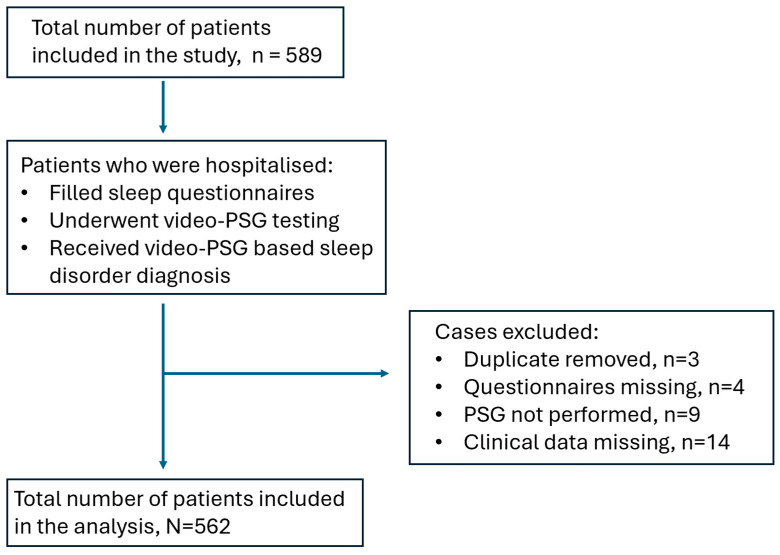
STROBE diagram.

**Figure 2 medicina-61-00780-f002:**
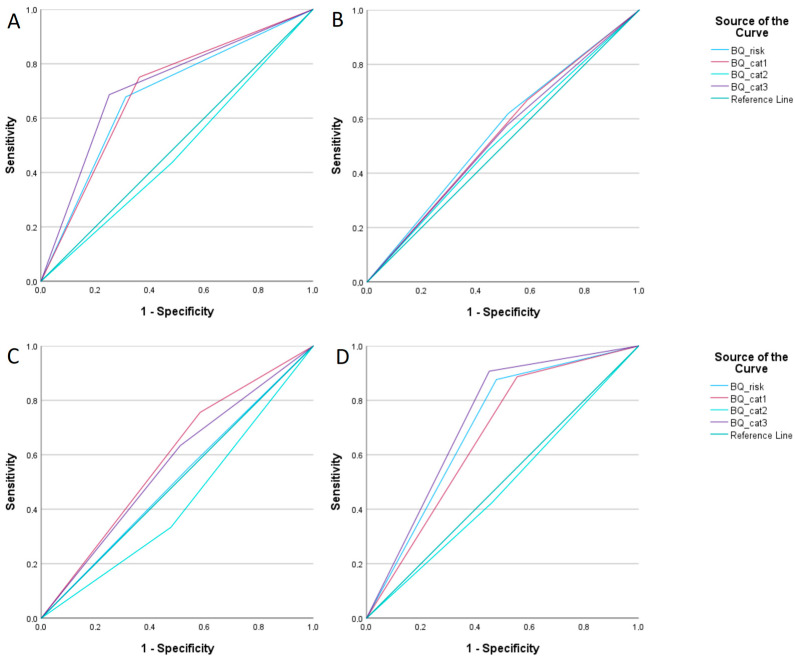
ROC curve analysis of Berlin Questionnaire for diagnosing obstructive sleep apnoea. (**A**)—Overall performance of BQ diagnosing any OSA. (**B**)—BQ for diagnosing mild OSA. (**C**)—BQ for diagnosing intermediate OSA. (**D**)—BQ for diagnosing severe OSA.

**Figure 3 medicina-61-00780-f003:**
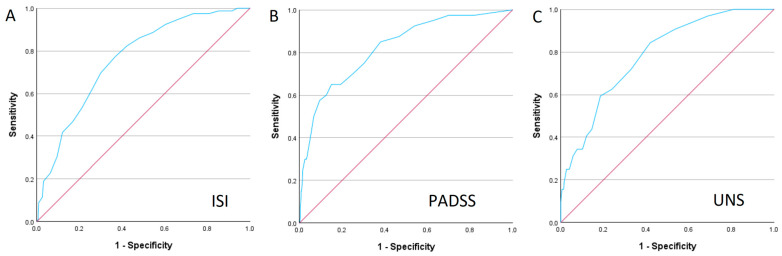
ROC curve analysis: (**A**) ISS for diagnosing insomnia; (**B**) PADSS for diagnosing parasomnia; (**C**) UNS for diagnosing hypersomnia.

**Figure 4 medicina-61-00780-f004:**
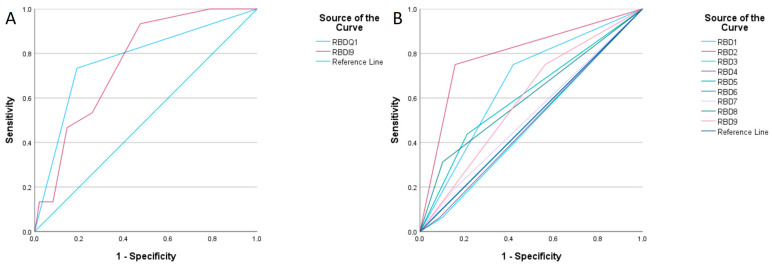
ROC curve analysis of RBD-I-9 and RBDQ1 for diagnosing REM sleep behaviour disorder. (**A**)—Comparison of both questionnaires. (**B**)—Comparison of each RBD-I-9 questions for diagnosing purposes of RBD.

**Figure 5 medicina-61-00780-f005:**
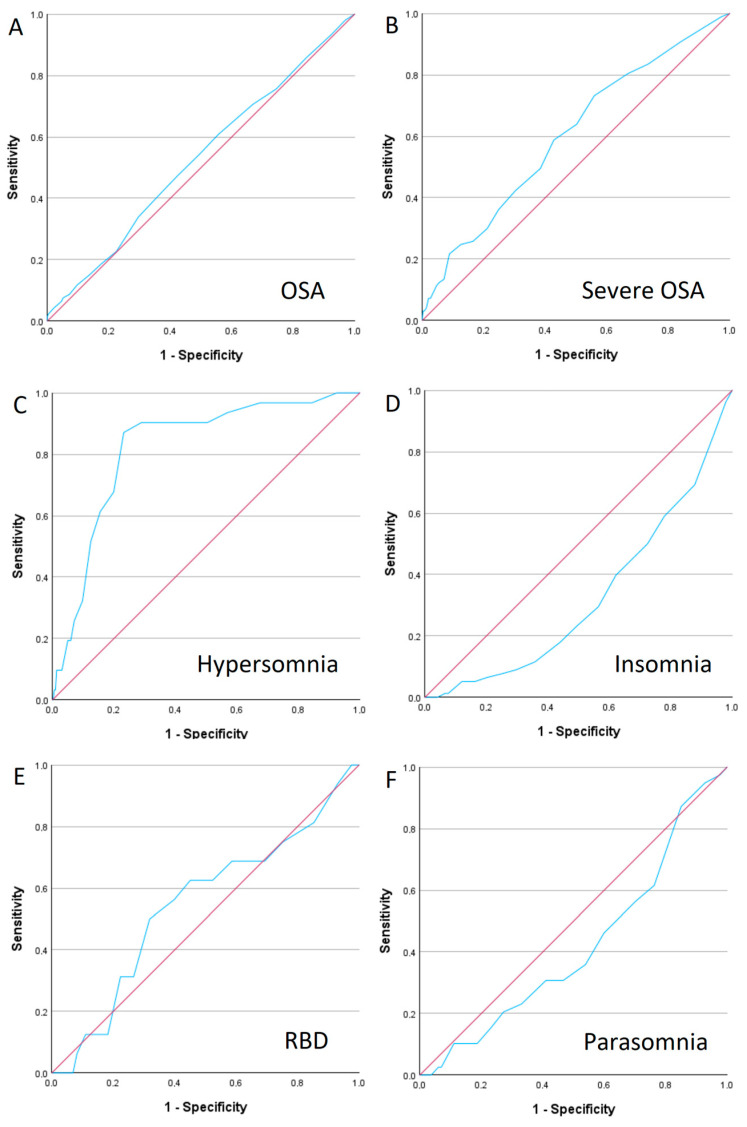
ROC curve analysis of Epworth Sleepiness Scale (ESS) for detecting various sleep disorders. (**A**)—Obstructive sleep apnoea (OSA); (**B**)—severe OSA; (**C**)—hypersomnia; (**D**)—insomnia; (**E**)—RBD; (**F**)—parasomnia.

**Table 1 medicina-61-00780-t001:** Demographic characteristics.

Sleep Disorder	OSA	Insomnia	Hypersomnia	Parasomnias	RBD
Prevalence	362, 64.4%	80, 14.2%	32, 5.7%	40, 7.1%	16, 2.8%
Gender	F 153, 42.3% M 209, 57.7% * *p* = 0.015	F 39, 48.8% M 41, 51.2%	F 24, 75.0% M 8, 25.9% * *p* < 0.001	F 19, 47.5% M 21, 52.5%	F 7, 43.8% M 9, 56.3%
Age, years	(−) 38.9 ± 15.5 (+) 53.03 ± 14.3 * *p* < 0.001	(−) 47.2 ± 16.0 (+) 52.6 ± 16.6 * *p* = 0.006	(−) 48.5 ± 16.3 (+) 40.2 ± 12.5 * *p* = 0.004	(−) 49.02 ± 15.7 (+) 34.5 ± 16.6 * *p* < 0.001	(−) 47.6 ± 16.2 (+) 59.2 ± 11.4 * *p* = 0.003
BMI, kg/m^2^	(−) 24.8 ± 4.2 (+) 30.6 ± 6.7 * *p* < 0.001	(−) 28.7 ± 6.8 (+) 27.6 ± 4.5	(−) 28.5 ± 6.3 (+) 29.1 ± 10.2	(−) 28.7 ± 6.4 (+) 26.9 ± 8.6	(−) 28.6 ± 6.5 (+) 26.5 ± 9.1
Comorbidities PAH	(−) 23, 13.7% (+) 145, 49.2% * *p* < 0.001	27, 43.5%	(−) 21, 84.0% (+) 4, 16.0% * *p* = 0.023	6, 20.7%	4, 30.8%
AF	(−) 3, 1.9% (+) 24, 8.2% * *p* = 0.008	5, 8.1	1, 4.0%	1, 3.4%	1, 7.7%
Diabetes	(−) 7, 4.6% (+) 32, 10.9% * *p* = 0.025	8, 12.9%	1, 4.2%	1, 3.4%	0
Dyslipidaemia	(−) 11, 7.0% (+) 59, 18.6% * *p* < 0.001	12, 17.4%	1, 3.8%	1, 4.3%	1, 7.1%
PD	6, 2.0%	0	0	1, 2.5%	8, 50.0%
Epilepsy	20, 6.8%	0	1, 4.0%	1, 3.4%	0
MS	1, 0.3%	0	0	0	0

OSA—obstructive sleep apnoea; RBD—REM sleep behaviour disorder; BMI—body mass index; PAH—primary arterial hypertension; AF—atrial fibrillation; PD—Parkinson’s disease; MS—multiple sclerosis; F—female; M—male; (+)—positive trait; (−)—negative trait. Results are expressed as absolute values and percentage, as well as mean ± standard deviation (SD) and *p* value (* only statistically significant values are presented).

**Table 2 medicina-61-00780-t002:** Overall performance of BQ for diagnosing OSA.

OSA	BQ	AUC	95% CI	Sensitivity	Specificity	*p*-Value
Mild	Overall	0.551	0.498–0.603	61.9%	48.2%	0.062
Intermediate	Cat. 1	0.586	0.524–0.648	75.6%	41.6%	0.01
	Cat. 2	0.428	0.365–0.492	33.3%	52.4%	0.032
	Overall	0.505	0.440–0.570	55.6%	45.5%	0.875
Severe	Cat. 1	0.667	0.614–0.719	88.7%	44.7%	<0.001
	Cat. 3	0.728	0.680–0.776	90.7%	54.9%	<0.001
	Overall	0.700	0.648–0.751	87.6%	52.3%	<0.001
Overall	Cat. 1	0.694	0.648–0.741	75.1%	63.8%	<0.001
	Cat. 3	0.718	0.427–0.527	68.6%	74.9%	<0.001
	Overall	0.683	0.637–0.730	67.8%	68.8%	<0.001

OSA—obstructive sleep apnoea; BQ—Berlin Questionnaire; AUC—area under the curve; 95% CI—95% confidence interval.

**Table 3 medicina-61-00780-t003:** Comparison of every question in RBD-I-9 scale and RBD1Q for diagnosing RBD.

RBD-I-9 Questions	AUC	95% CI	*p*-Value	Sensitivity	Specificity
**Question 1**	**0.665**	**0.538–0.793**	**0.024**	**75.0%**	**58.1%**
**Question 2**	**0.796**	**0.671–0.921**	**<0.001**	**75.0%**	**84.2%**
Question 3	0.481	0.342–0.621	0.799	-	-
Question 4	0.487	0.346–0.628	0.863	-	-
Question 5	0.613	0.462–0.764	0.124	-	-
Question 6	0.501	0.357–0.645	0.986	-	-
Question 7	0.519	0.372–0.666	0.797	-	-
Question 8	0.605	0.449–0.762	0.151	-	-
Question 9	0.594	0.461–0.727	0.202	-	-
**Overall RBDI9**	**0.758**	**0.665–0.851**	**0.001**	**93.3%**	**52.5%**
**RBD1Q**	**0.772**	**0.641–0.903**	**<0.001**	**73.3%**	**81.0%**

RBD-I-9—Innsbruck REM Sleep Behaviour Disorder Inventory 9-item scale; AUC—area under the curve; 95% CI—95% confidence interval.

## Data Availability

All data are available on reasonable request to the corresponding author.

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
