# Peer review of "Can We Reduce the Diagnostic Burden of Sleep Disorders? A Single-Centre Study of Subjective and Objective Sleep-Related Diagnostic Parameters"

_medicina, 2025, doi:10.3390/medicina61050780_

Round 1
Reviewer 1 Report
Comments and Suggestions for Authors
Thank you for allowing me to review the article titled “Can we reduce the diagnostic burden of sleep disorders? A single-centre study of subjective and objective sleep-related diagnostic parameters”
- Title: Include the country or region in the title to clearly reflect the geographic context of the study, as mentioned in the introduction, methods, and discussion sections.
- Abstract: It is recommended to include key results in the abstract, such as the sensitivity and specificity of the diagnostic questionnaires compared to the reference method.
- Introduction: It is recommended to clearly state the objective of the study in the last paragraph of the introduction.
- Introduction: It is recommended to include a more detailed review of previous studies comparing subjective questionnaires with polysomnography, to better highlight the gap in the literature that this study aims to address.
- Methods: It is recommended to provide a clear and detailed description of the study design. The population inclusion and exclusion criteria should be clearly defined, and the instruments used should be described in detail, including their validity and reliability. In addition, a description of the statistical analysis methods should be included.
Results:
- Present the results clearly, using tables and figures to summarize the key findings. Include key metrics such as sensitivity, specificity, and comparisons with polysomnography, along with a brief description of the statistical methods used.
- Provide more detailed information on the statistical analysis performed, including p-values, confidence intervals, and subgroup analysis.
- Interpret the p-values presented in the tables, focusing on statistically significant items, and explain their relevance to the detection of sleep disorders.
- Include a brief interpretation of key metrics such as sensitivity and specificity.
- Discussion: It is recommended to expand the discussion by interpreting the results in the context of existing literature, emphasizing their clinical implications. Limitations, such as potential biases or sample size issues, should be addressed to provide a more comprehensive understanding of the study's findings. Additionally, future research directions should be proposed, particularly exploring the factors that influence the accuracy of subjective questionnaires in diagnosing sleep disorders and analyzing their potential applications in clinical practice.
- The conclusions are very brief.
Author Response
Good evening,
Thank you for your comments. Please, find our reply to in the added word file.
Best regards,
Article writing team

Reviewer 2 Report
Comments and Suggestions for Authors
The manuscript offers valuable insights into its field, and I commend the authors for their efforts. Reducing patient burden and avoiding futile screening are the primary concerns in the contemporary approach to sleep-related breathing disorders. The authors adequately discuss these issues and provide a plausible explanation for the lower performance of certain questionnaires in their research. They propose a strategic method aimed at simplifying and optimizing the screening process in various clinical settings for sleep-related disorders.
Certain minor revisions are needed for the paper to enhance clarity and advance through the review process.
The introduction is well-written and presents a logical flow of ideas, effectively setting the stage for the article. In Lines 73-79, the authors are encouraged to clarify and define the study outcomes more precisely, as they are not clearly delineated in the manuscript.
At Line 126, ICSD-3 should include a reference.
To enhance the validity of the research findings, a suggestion is to include in the Discussion section a paragraph that compares the authors’ sample size with those used in previous studies testing the sensitivity and specificity of different screening tests for OSA (e.g., at Lines 376-381 – ESS for OSA). This comparison could provide helpful context for readers regarding the implications of the study findings and the reliability of the results.
Minor text editing details, such as references in square brackets, are also suggested.
Author Response
Good evening,
Thank you for your comments. Please, find our reply to your comments in the added word file.
Best regards,
Article writing team

Round 2
Reviewer 1 Report
Comments and Suggestions for Authors
Accept in present form